# Multi-Omic Approaches in Cancer-Related Micropeptide Identification

**DOI:** 10.3390/proteomes12030026

**Published:** 2024-09-13

**Authors:** Katarina Vrbnjak, Raj Nayan Sewduth

**Affiliations:** VIB-KU Leuven Center for Cancer Biology (VIB), 3000 Leuven, Belgium

**Keywords:** non-canonical peptides, cancer, SEPs, micropeptides, multi-omics, peptide drugs, microproteome complexity

## Abstract

Despite the advances in modern cancer therapy, malignant diseases are still a leading cause of morbidity and mortality worldwide. Conventional treatment methods frequently lead to side effects and drug resistance in patients, highlighting the need for novel therapeutic approaches. Recent findings have identified the existence of non-canonical micropeptides, an additional layer of the proteome complexity, also called the microproteome. These small peptides are a promising class of therapeutic agents with the potential to address the limitations of current cancer treatments. The microproteome is encoded by regions of the genome historically annotated as non-coding, and its existence has been revealed thanks to recent advances in proteomic and bioinformatic technology, which dramatically improved the understanding of proteome complexity. Micropeptides have been shown to be biologically active in several cancer types, indicating their therapeutic role. Furthermore, they are characterized by low toxicity and high target specificity, demonstrating their potential for the development of better tolerated drugs. In this review, we survey the current landscape of known micropeptides with a role in cancer progression or treatment, discuss their potential as anticancer agents, and describe the methodological challenges facing the proteome field of research.

## 1. Introduction

Cancer remains a major public health problem globally, in spite of the strides made in its early diagnosis and treatment. Recent data show that cancer is the leading cause of premature mortality in the world, a trend that is expected to increase in the coming years [1]. This increasing incidence of malignant disease can be attributed to various causes, including an aging population, and lifestyle factors such as tobacco use or physical inactivity. One of the main challenges in successful cancer treatment is the development of drug resistance, a phenomenon where cancer cells become less responsive or completely unresponsive to therapeutic agents. This resistance can be innate or acquired, and can occur through various mechanisms, such as increased drug efflux, drug inactivation, or signaling pathway alteration [2]. Another challenge closely tied with effective cancer therapy is the toxicity exhibited by many chemotherapeutic drugs. Since the cytotoxic effect of most anticancer drugs is not limited only to cancer cells, side effects such as kidney toxicity in cisplatin or liver toxicity in mercaptopurine are common [3,4]. Cancer immunotherapy also frequently leads to unique toxicity profiles that are distinct from the profiles of chemotherapy drugs, and thus require specific management [5]. It is therefore critical to explore novel strategies in cancer therapy, focusing on approaches that limit the possibility of resistance and toxicity.

The capacity of the human genome to produce functional proteins has been historically underestimated, as the proteome complexity is still not fully understood. One of the major limitations to deciphering the coding capacity of the cell has been due to an arbitrary assumption that proteins composed of fewer than 100 amino acids are unlikely to be translated and functional. Additionally, bioactive peptides have mostly been studied in the form of cleavage products of larger precursor proteins. However, omics-based technologies such as ribosome profiling, bioinformatics methods, and improved mass spectrometry have given researchers critical insights into this overlooked part of the translatome, identifying proteoforms coded for by parts of the genome incorrectly labeled as junk, such as intronic sequences, non-coding RNA, and pseudogenes [6] (Figure 1). By combining these approaches, researchers have been able to define more closely which open reading frames code for functional proteins, as opposed to simply occurring in the genome randomly. This gave rise to synergistic strategies such as proteogenomics to better analyze the available data and detect these small and often low-abundance proteins, further highlighting the microproteome landscape [7]. By identifying and describing these previously ignored protein species, the complexity of the human microproteome will begin to be understood more fully, resulting in an enhanced ability to identify novel therapeutic targets.

Non-coding RNA processing is known to be dysregulated in cancer and associated with a plethora of genomic alterations, and evidence is emerging about small ORFs encoded by them that code for micropeptides. Similarly, functional studies have demonstrated that a large number of these non-canonical micropeptides are dysregulated and have an important role in a multitude of cancer-related biological processes, including carcinogenesis and cancer cell survival [8,9,10]. Several of these micropeptides present a tumor-suppressive effect through their interaction with oncogenic signaling pathways, while the dysregulation of other micropeptides can be a driver of oncogenesis, further highlighting the role of these small proteins in cancer [11]. There is ample evidence for micropeptide involvement in many aspects of the cell machinery, in particular those that are crucial for tumor emergence and progression, such as metabolic regulation within the mitochondria and mRNA expression regulation within the nucleus [12]. These findings will pave the way to a variety of novel small molecules revealed by the landscaping of the microproteome complexity, which will be of use in the continuous fight against cancer.

This review aims to give a comprehensive insight into the methods employed in the prediction and detection of micropeptides, in particular, novel multi-omic methods that represent the cutting edge of peptidomic research. We also bring into focus their underutilized potential as specific and attractive anticancer agents.

## 2. Multi-Omic Approaches for Micropeptide Identification and Detection

In order to fully understand the translational and consequently therapeutic potential of micropeptides, a multi-omics approach integrating genomics, transcriptomics, and proteomics data is essential (Table 1). This sort of approach allows for a comprehensive analysis of micropeptides and their interactions in a complex biological sample (Figure 2).

One of the most used techniques in micropeptide detection is ribosome profiling, also known as Ribo-Seq or ribosome footprinting. Developed by Ingolia et al. in 2009, this deep-sequencing-based omics technique utilizes deep sequencing to identify fragments of mRNA that are bound to ribosomes and therefore actively translated, using three-nucleotide periodicity as proof of active translation as opposed to random ribosome binding events. The technique is also valued for its possibility to identify peptides that contain non-traditional start codons, which is a common occurrence in non-canonical micropeptides [13,14]. This technique has given rise to various methods that aim to improve its reliability, such as Poly-Ribo-Seq, which profiles polysomal fractions instead of all ribosomal-bound mRNAs. This approach provides data for mRNAs that are bound by multiple ribosomes and distinguishes them from sporadic translational events, giving a robust insight into actively translated portions of the genome and lowering translational noise [15]. Ribo-Seq data are frequently combined with other omics techniques to improve reliability. This combination can be made with transcriptomics (mRNA sequencing) and mass spectrometry, which has been shown to increase the predictive value of Ribo-Seq [16]. Through combining Ribo-Seq results with bioinformatic pipelines, in particular, when combined with extensive sequencing depth and increased resolution of framing, researchers have been able to reliably detect lncRNA-encoded peptides [17]. When combined with single-cell transcriptomics, an analysis of translational dynamics down to a resolution of a single live cell has been achieved, significantly increasing the sensitivity and scalability of Ribo-Seq. [18] Limitations of this technique have been identified, such as the complexity of the produced data, which can be challenging to interpret, often inaccessible costs, sensitivity to experimental conditions, and a tendency to miss low-abundance transcripts. Furthermore, Ribo-Seq cannot detect or predict protein stability and degradation, folding, or post-translational modifications (PTMs) of micropeptides [19,20].

Out of the available proteomics methods that can provide evidence for a direct detection of peptides, mass spectrometry (MS) remains the most popular. This technique allows for the identification and quantification of proteins in a biological sample, and works by ionizing proteins and giving a measurement of the mass-to-charge ratio of the resulting ions [21]. Tandem mass spectrometry (MS/MS) and liquid chromatography followed by tandem mass spectrometry (LC-MS/MS) have been performed in the past to validate the existence of non-canonical micropeptides [22]. An important part of analyzing MS data in the context of micropeptidome research is using an appropriate library, as reference databases of known proteins such as UniPROT will only contain canonical proteins. To this end, proteogenomics uses proteomic data to re-annotate parts of the genome that have previously been annotated as non-coding, providing a database in which micropeptide spectra may be found instead of using traditional reference databases [23]. A database of peptides to use for this purpose may be acquired from ribosome profiling data, and it has been shown that this approach results in an expanded protein identification rate [24]. MS has been successfully combined with RNAseq data to provide a more comprehensive overview of protein-coding regions within long non-coding RNA [25,26]. Micropeptides have also been identified through MS when using a three-frame translated lncRNA database as a reference, suggesting that many lncRNAs are indeed translated [27]. Although MS for the direct detection of micropeptides is possibly the most valuable evidence for micropeptide translation, researchers face unique challenges with this approach, as micropeptides are commonly considered unsuitable for traditional MS for a variety of reasons. For instance, the dynamic range of MS instruments may not be sufficient to capture these low-abundance and small peptides [28], and the preparation of samples for MS often carries a risk of loss of tiny proteins or their rapid degradation. Trypsin digestion in preparation for MS may also not be appropriate for micropeptides, since their prohibitively small size often does not provide enough cleavage sites for trypsin to act on. For this purpose, alternative proteases such as Glu-C, LysN, or Asp-N can be used.

Bioinformatics approaches focused on small ORF discovery have also long struggled with finding appropriate algorithms. Standard computational methods can easily fail when trying to detect micropeptides for multiple reasons, including their short length, high abundance, and lack of conservation. The usual algorithms rely on an ORF’s amino acid or nucleotide sequence to determine a putative protein’s evolutionary conservation or similarity to previously identified proteins. This approach is normally optimized for longer proteins, which is why it frequently fails for ORFs smaller than 80 amino acids (aa). Another issue is that micropeptides are in many cases not evolutionarily conserved, being relatively young gene products [29]. However, recent advancements in machine learning and artificial intelligence have yielded new methods to use for the search for micropeptides. These novel tools harness machine learning to predict which sequences may be translated, using features intrinsic to the ORF sequence. Some of these features include the DNA Fickett score, transcript length, stop codon frequency, and GC content. Within the peptide itself, these features include its molecular weight, isoelectric point, grand average of hydropathicity (GRAVY) score, and instability index [30]. Although the reliability of computational methods to determine coding transcripts has improved dramatically in the past few years, to achieve a fuller understanding of the protein-coding capacity of a given dataset, it is still important to combine it with wet-lab methods. Ribo-Seq is a popular choice, and ORF-finding pipelines will frequently include an overlap step with Ribo-Seq data to further validate their algorithm [31]. Machine learning has been successfully utilized in the past to predict a large amount of ORFs from DNA sequences [32]. Deep learning, in particular neural network architectures, has also been used to analyze Ribo-Seq data [33], and deep learning-based frameworks have been employed to study translation elongation in these datasets [34]. When combining bioinformatics approaches with mass spectrometry, researchers have successfully used machine learning models to impute missing values in proteomics data and to solve highly multiplexed spectra [35], which suggests that these advanced computational tools may be indispensable in the analysis of MS results when dealing with micropeptides, given their low abundance and high propensity for degradation. Artificial intelligence algorithms can process vast amounts of spectral data, identifying patterns and correlations that are not easily discernible through traditional methods.

Other novel omics methods have been utilized to identify and analyze micropeptides in the past. There have been recent efforts to utilize spatial omics, an emerging technique that provides a three-dimensional outlook into the molecules present in a given tissue, which could provide an important spatial context for these small proteins. Ribosome profiling has been combined with spatial omics in a method named ribosome-bound mRNA mapping, which allows for a spatially-resolved insight into protein synthesis at a single-cell resolution [36]. Spatial omics have also been used to comprehensively investigate mouse tissue through combining RNA sequencing with ribosome profiling, an approach that has yielded insights into pervasive lncRNA translation and uncovered many novel micropeptides [37]. Transcriptomics is widely used in the field, with RNA sequencing being essential for the identification and expression profiling of sORFs to determine which may be translated. This technique uses high-throughput sequencing to identify the quantities of RNA in a given sample, providing a comprehensive insight into the transcriptome of a biological sample. RNA sequencing helps to reclassify transcripts previously classified as non-coding by providing evidence of their coding potential [38]. RNA-seq can be used independently of other techniques, but it is frequently combined with ribosome profiling and other more direct micropeptide-identifying methods, since RNA expression and transcription do not necessarily imply translation [39]. HLA immunopeptidomics has been used to identify non-canonical ORFs, a method for directly profiling the peptides presented by the human leukocyte antigen system through immunoaffinity purification and LC-MS/MS [40]. Concerning proteomics, micropeptides may be easier to detect using a targeted approach, such as data-independent acquisition (DIA) when performing MS. It has been shown that DIA has a superior ability to quantify micropeptides when compared to a traditional data-dependent approach, although identification is only possible with prior knowledge of targets [41]. To validate the translation and function of a micropeptide, a CRISPR-Cas9 system can be utilized to precisely target these small ORFs in the genome, and researchers can endogenously tag them for detection or examine the loss-of-function phenotype upon micropeptide knockout [42]. Immunodetection of micropeptides may be challenging because of their low expression, combined with the difficulty of epitope development for such a small protein, but through such methods as tag knock-in via CRISPR-Cas9, this becomes more accessible.

**Table 1 proteomes-12-00026-t001:** List of multi-omic approaches for micropeptide identification and detection.

Techniques	Details	Limitations/Challenges
Ribo-Seq [13,14,16,17,18]	Identifies mRNA fragments bound to ribosomes, detects non-traditional start codons, combined with other omics techniques for reliability	Complexity of data, cost, sensitivity to conditions, misses low-abundance transcripts, cannot detect protein stability, degradation, folding, or PTMs
Poly-Ribo-Seq [15]	Profiles polysomal fractions, distinguishes mRNAs bound by multiple ribosomes, reduces translational noise	Technical complexity, cost, sensitivity
Mass Spectrometry (MS) [21,22,23,24,25,26,27,28]	Identifies and quantifies proteins, ionizes proteins and measures mass-to-charge ratio, uses appropriate libraries for micropeptide detection	Dynamic range limitations, risk of protein loss or degradation, trypsin digestion issues
Machine Learning [30,31,32,33,34,35]	Predicts ORFs, uses features such as DNA Fickett score, transcript length, stop codon frequency, GC content, molecular weight, isoelectric point, GRAVY score, instability index	Struggles with short length of ORFs and lack of conservation of micropeptides, prediction algorithm quality may vary
Spatial Omics [36,37]	A spatial insight into protein synthesis at a single-cell and subcellular resolution	The integration of diverse data modalities, computational analysis, and the standardization of protocols
RNA Sequencing for sORF Identification and Expression Profiling [38,39]	Used for the identification and expression profiling of short open reading frames (sORFs)	Issues with detection of small ORFs, RNA expression does not imply translation
CRISPR-Cas9-Based Methods [42]	Can be used to precisely target and edit specific genetic sequences	Risk of off-target effects, changes in stability after tagging

## 3. Non-Canonical Micropeptides in Cancer

### 3.1. Cancer-Suppressing Micropeptides

Non-canonical micropeptides may be encoded by many regions of the genome currently not known for their coding capacity, such as intergenic regions, antisense RNAs, microRNAs (miRNAs), circular RNAs (circRNAs), small nucleolar RNAs (snoRNAs), and pseudogenes. Most non-coding RNA translation research focuses on lncRNAs, miRNAs, and circRNAs, citing their translation potential and ability to code for functionally significant micropeptides [43] (Table 2). For instance, a non-canonical peptide named MIAC has recently been shown to inhibit renal cell carcinoma progression. This micropeptide, encoded by lncRNA AC025154.2, gained recognition by being down-expressed in renal cancer tissue according to The Cancer Genome Atlas (TCGA) data, and it was found that it can inhibit the development of renal cancer in vitro and in vivo. The mechanism of its action was uncovered through high-throughput transcriptome sequencing, and it was found that it inhibits EGFR expression and its associated downstream signaling pathway. Moreover, it was found that a chemically synthesized MIAC polypeptide can also inhibit renal cancer in vitro and in vivo, highlighting its potential in renal cancer therapy [44]. The same micropeptide is also known to suppress head and neck squamous cell carcinoma proliferation and metastasis through its interaction with AQP2 and regulation of the SEPT2-ITGB4 axis [45]. Another lncRNA, LINC00665, is known to encode for a micropeptide named CIP2A-BP. This micropeptide was found through a bioinformatics analysis of Ribo-Seq and RNA-seq datasets, and further verified by polysome profiling. Functional tests showed that the micropeptide, but not its lncRNA, inhibits the migration and invasion of triple-negative breast cancer cells in vitro and in vivo. CIP2A-BP was also directly administered into mice, and it was shown that this approach can significantly suppress breast cancer metastasis and invasion [46]. The lncRNA HOXB-AS3 has been found to encode for a 53-aa peptide of the same name through ribosome profiling. Low expression of this peptide was shown to be correlated with a poor prognosis in colorectal carcinoma patients. The peptide was then found to suppress cancer cell growth, colony formation, migration, and invasion in vitro as well as tumorigenesis in vivo by binding to heterogeneous nuclear ribonucleoproteins A1 and suppressing aerobic glycolysis [47].

A 99-aa micropeptide named KRASIM encoded by lncRNA NCBP2-AS2, discovered through ribosome profiling and notably differently expressed in normal hepatocytes and hepatocellular carcinoma (HCC) cells, has been shown to interact with KRAS and inhibit ERK signaling in HCC cells. This ultimately suppresses HCC growth and proliferation [48]. An example of circular RNAs coding for micropeptides with cancer relevance is PINT87aa, a peptide coded for by a circRNA formed by exon 2 of LINC-PINT after circularization. This micropeptide has been shown to induce cellular senescence in hepatocellular carcinoma cells. The mechanism was elucidated by structural analysis and co-immunoprecipitation, and the researchers found that the micropeptide binds to the DNA-binding domain of FOXM1, inhibiting PHB2 transcription and promoting cellular senescence [49]. A micropeptide named AF127577.4-ORF, encoded by its namesake lncRNA, has been shown to diminish glioblastoma cell proliferation through its regulation of METTL3 and the ERK pathway, which could be a promising approach to glioblastoma management [50]. A 133-aa microRNA-encoded peptide named miPEP133 has been identified as a tumor-suppressor microprotein in ovarian cancer cell lines [51]. In a recent study, non-canonical micropeptides have been discovered to make up a major source of tumor-specific antigens in liver cancer. In addition, 117 hepatocellular carcinoma patients’ samples yielded novel cancer antigens coded for by tumor-specific long noncoding RNAs [52]. A polypeptide coded for by lncRNA, ASRPS, was shown to inhibit triple-negative breast cancer angiogenesis through its interaction with STAT3 and subsequent downregulation of VEGF [53]. In glioblastoma, mitochondrial quality control has been found to be disrupted by a micropeptide named MP31. This small protein, encoded by an upstream ORF located in the 5′UTR of PTEN, was shown to suppress mitochondrial activity in glioblastoma, leading to mitochondrial damage and ROS accumulation in cells. It was also shown to act selectively, enhancing glioblastoma cell sensitivity to chemotherapy, but with no effect on microglia and astrocytes, thus showing its potential in targeted cancer therapy [54]. A lncRNA-encoded micropeptide named TINCR that is selectively expressed in the epithelium has been shown to exhibit tumor suppressor qualities in squamous cell carcinoma. When this micropeptide is overexpressed in SCC cells, cancer growth both in vitro and in vivo is suppressed, and the loss of expression of this protein is shown to be a prognostic marker for lower overall survival and more frequent metastases [55].

**Table 2 proteomes-12-00026-t002:** Biological mechanisms and therapeutic implications of previously identified cancer-suppressing micropeptides.

Micropeptide	Encoded by	Function	Mechanism	Cancer Type
MIAC [44]	lncRNA AC025154.2	Inhibits renal cell carcinoma progression	Inhibits EGFR expression and downstream signaling	Renal cancer
MIAC [45]	lncRNA AC025154.2	Suppresses head and neck squamous cell carcinoma proliferation and metastasis	Interacts with AQP2 and regulates SEPT2-ITGB4 axis	Head and neck cancer
CIP2A-BP [46]	lncRNA LINC00665	Inhibits migration and invasion of triple-negative breast cancer cells	binds CIP2A to inhibit PI3K/AKT/NFkB pathway	Breast cancer
HOXB-AS3 [47]	lncRNA HOXB-AS3	Suppresses cancer cell growth, colony formation, migration, and invasion	Binds to heterogeneous nuclear ribonucleoproteins A1, suppresses aerobic glycolysis	Colorectal cancer
KRASIM [48]	lncRNA NCBP2-AS2	Inhibits ERK signaling	Interacts with KRAS	Hepatocellular carcinoma
PINT87aa [49]	circRNA formed by exon 2 of LINC-PINT	Induces cellular senescence	Binds to DNA-binding domain of FOXM1, inhibits PHB2 transcription	Hepatocellular carcinoma
AF127577.4-ORF [50]	lncRNA AF127577.4	Diminishes glioblastoma cell proliferation	Regulates METTL3 and ERK pathway	Glioblastoma
miPEP133 [51]	precursor of miR-34a	Tumor-suppressor	Interacts with HSPA9 to disrupt its function	Ovarian cancer
ASRPS [53]	lncRNA ASRPS	Inhibits angiogenesis	Interacts with STAT3, downregulates VEGF	Breast cancer
MP31 [54]	uORF in 5′UTR of PTEN	Causes mitochondrial damage in glioblastoma cells	Competes with V-ATPase A1 for lactate dehydrogenase B binding	Glioblastoma
TINCR [55]	lncRNA TINCR	Tumor-suppressor	TP53 target gene	Squamous cell carcinoma
AC115619–22aa [56]	lncRNA AC115619	Represses cancer growth and metastasis	Binds to WTAP to disrupt its function	Hepatocellular carcinoma
SMIM26 [57]	lncRNA LINC00493	Exhibits anti-metastatic activity, maintains mitochondrial activation	Interacts with AGK to deactivate AKT signaling, interacts with SLC25A11	Clear cell renal cell carcinoma
MPM [58]	lncRNA LINC00116	Inhibits hepatocellular carcinoma metastasis	Regulates the activity of mitochondrial complex 1 through its interaction with NDUFA7	Hepatocellular carcinoma

The lncRNA AC115619 codes for a conserved 22-aa peptide AC115619–22aa, identified through bioinformatics methods and Ribo-Seq and subsequently validated in hepatocellular carcinoma cells. It was shown that this peptide exhibits anticancer qualities through its interaction with WTAP. This peptide was shown to directly inhibit cancer growth in hepatocellular carcinoma when the synthetic peptide was injected into xenografts, highlighting its potential as a therapeutic agent [56]. A 95-aa micropeptide SMIM26, encoded by LINC00493, has an important effect in clear cell renal cell carcinoma anti-metastatic activity. It was discovered to interact with AGK to deactivate AKT signaling, suppressing cancer metastasis, and also to maintain mitochondrial activation through its interaction with SLC25A11. Since this peptide is consistently downregulated in this type of cancer, the authors suggest that its restoration may serve as an option for renal cell carcinoma treatment [57]. MPM (micropeptide in mitochondria), also known as MOXI/MTLN, has recently been linked to cancer through its role in hepatocellular carcinoma metastasis. This 56-aa conserved mitochondrial peptide was shown to regulate the activity of mitochondrial complex 1 through its interaction with NDUFA7, and the authors have also demonstrated that its downregulation promotes hepatocellular carcinoma metastasis, suggesting a possible therapeutic use [58].

Overall, the large amount of evidence to support the role of non-canonical micropeptides in the suppression of cancer should not be ignored. These peptides show great promise for the development of anticancer vaccines and targeted therapy, especially when their cost-effectiveness, versatility, ease of synthesis and modification, high specificity, and low toxicity are taken into account.

### 3.2. Cancer-Promoting Micropeptides

Just as there are numerous micropeptides that act as anticancer agents, many have been reported to drive oncogenesis and cancer progression (Table 3). ASAP, a 94-aa non-canonical micropeptide encoded by LINC00467, was implicated in mitochondrial metabolism and ATP production in colorectal cancer (CRC). It binds to ATP synthase inside the mitochondria and promotes its activity, ultimately driving CRC cell proliferation and leading to a worse outcome. The authors used a CRISPR/Cas9 system to disrupt this micropeptide and showed that its knockout suppresses the growth of CRC xenografts in vivo [59]. In hepatocellular carcinoma, the highly expressed lncRNA LINC00998 was found to code for a conserved 59-aa micropeptide named SMIM30. Among its protein interactors, SRC/YES1 were found, and it was revealed that SMIM30 has an important role in MAPK signaling pathway activation through this interaction, thus driving cancer proliferation and growth [60]. Furthermore, in hepatocellular carcinoma tumor metastasis, a micropeptide named STMP1 was found to interact with DRP1 in the mitochondria and promote tumor cell migration [61], whereas another micropeptide named JunBP was found to bind to c-Jun and promote its activation, thus driving cancer metastasis [62]. XBP1SBM, a 21-aa micropeptide encoded by the lncRNA MLLT4-AS1, promotes the growth, angiogenesis, and metastasis of triple-negative breast cancer through the XBP1s/VEGF pathway [63]. In clear cell renal cell carcinoma, the non-canonical peptide ACLY-BP drives cell proliferation by enhancing the stability of ACLY and leading to both increased acetyl-CoA production and lipid deposition [64]. APPLE, a micropeptide encoded by ASH1L-AS1, interacts with the PABPC1 complex to initiate protein translation, and in particular, a specific subset of mRNAs critical for tumorigenesis. The authors show that silencing of this micropeptide slows leukemic cancer cell proliferation and infiltration [65].

The non-canonical translation of the micropeptide ASNSD1-uORF has been shown to be necessary for childhood medulloblastoma cell survival through its interaction with the prefoldin-like chaperone complex, presenting an attractive target for medulloblastoma therapy [66]. In ovarian cancer, a lncRNA-encoded micropeptide called DDUP was linked to cisplatin resistance. This peptide was found to have higher expression levels in cells after DNA damage, and its upregulation was subsequently linked to therapy resistance in ovarian cancer through its mediation of DNA damage repair. When this peptide was knocked out via CRISPR/Cas9, the authors observed a cisplatin-sensitive phenotype in the cells, suggesting a new strategy for treatment-resistant ovarian cancer [67]. PACMP, a 44-aa lncRNA-encoded micropeptide discovered in breast cancer, has shown a role in both cancer cell growth and cancer therapy resistance. The authors extensively outline its role in double-strand DNA break repair and show that targeting this micropeptide confers a synthetic lethal effect to tumor cells, as well as sensitizing breast cancer xenografts to multiple chemotherapeutic agents [68]. Also in breast cancer, antisense strand-encoded micropeptide TRPC5OS was found to induce tumorigenesis through its interaction with ENO1. The peptide is upregulated in breast cancer cell lines and tissue, and its overexpression significantly increases cancer cell survival in vitro. The authors suggest that TRPC5OS expression is a potential prognostic index for breast cancer [69].

Lymphatic node metastasis of gastric cancer has also been linked to micropeptides, with the exo-lncRNA-encoded pep-AKR1C2 being shown to promote lymphangiogenesis and metastasis. The peptide was identified via Ribo-Seq, validated in gastric cancer cells, and shown to promote CPT1A expression in lymphatic endothelial cells, leading to higher levels of fatty acid oxidation and promoting metastasis. This effect was observed both in vitro and in vivo, hinting at a new target for gastric cancer therapy [70]. ATMLP, a 90-aa lncRNA-encoded peptide, has been shown to be implicated in non-small cell lung cancer malignancy. High expression levels of this peptide correlate with a poor survival for patients, and when its function was more closely examined, it was found to promote tumorigenesis of epithelial cells through its interaction with NIPSNAP1. Remarkably, the levels of ATMLP in the serum increase as the tumor progresses, making it a sensitive diagnostic biomarker for non-small cell lung cancer [71].

These tumor-promoting micropeptides represent a large array of potential therapeutic targets and prognostic biomarkers that can be used in cancer treatment. Targeting these peptides may prove to be a promising strategy for fighting this disease, either through the application of specific inhibitors, antisense oligonucleotides, or RNA interference.

## 4. Discussion

Given the complex roles that micropeptides play in fine-tuning the molecular mechanisms of cancer, they constitute attractive targets for cancer therapy. Considering the large number of tumor-suppressing micropeptides that reveal the complexity of the microproteome, there is undeniable evidence that this class of proteoforms could provide novel and as-of-yet unexplored avenues for cancer therapy development. The tissue-specificity of micropeptides enables them to be used as biomarkers, which can allow researchers to accurately and reliably determine a cancer subtype [72]. When using micropeptides to target specific cellular pathways or proteins, their high specificity allows them to act with high precision, thereby potentially reducing the off-target effects and toxicity that are so common in traditional cancer therapy. Micropeptide therapy could be used in conjunction with other cancer therapies such as radiotherapy or immunotherapy to enhance the efficacy of both, leading to a possible synergistic effect. The ubiquity of drug and therapy resistance in cancer has a clear hope of being circumvented through targeting these small proteins and using them for therapeutic purposes. Their small size, generally not exceeding 100aa, allows for ease of chemical synthesis, thus minimizing the costs involved in their production. They can also easily be modified to be more structurally stable through approaches such as chemical stapling, or coupled with delivery methods such as nanoparticles for greater stability. Because of their specificity, micropeptides are sources of targetable tumor-specific antigens, which is key for developing new tumor vaccines [73,74,75,76]. This has the potential to revolutionize precision medicine and cancer therapy, making the treatment of tumors both more economical and less systemically toxic.

There, however, remain many challenges in effectively utilizing micropeptides in cancer therapy. Given all the difficulties in detecting, validating, and investigating their mechanisms due to the complex nature of the microproteome, researchers are still struggling to understand the function of many non-canonical micropeptides, which complicates their development as cancer therapeutics. Since they tend to be susceptible to degradation, large-scale production may be difficult and bioavailability may be low. Peptide-based drugs are known to have a short circulating half-life and are rapidly cleared from the organism, as well as having poor oral bioavailability and limited permeability across biological membranes [77]. There have been improvements in peptide-based drug delivery systems, such as the use of biodegradable particles as carriers and cell-penetrating peptides for precision delivery [10,78,79], which may help circumvent some of these challenges. Seeing how non-canonical micropeptides have only recently emerged as important players in cell biology, we have yet to fully understand their potential, and much work remains to be done in this field.

Many novel methods have been developed to capture and understand these small proteins, such as ribosome and polysome sequencing, improved mass spectrometry with custom reference databases, and bioinformatics methods with an emphasis on machine learning algorithms. When considering all the challenges linked with successful micropeptide identification, classification, and analysis, it is vital to employ a comprehensive and systematic multi-omic approach. Combining several omics methods and using a diverse palette of techniques may be the key to uncovering the hidden realm of non-canonical micropeptides and revealing the complex landscape of the microproteome.

Recent findings have finally begun to shed light on the actual complexity of the human proteome by investigating these historically ignored short ORFs that are present in the parts of the genome annotated as non-coding. There have been many recent publications reviewing these exciting new developments in the world of proteomics, such as Yuanyuan 2022 [80], Vitorino 2021 [81], Tharakan 2021 [82], and Lu 2023 [83]. These reviews all offer a comprehensive insight into the methods used in micropeptide research and provide plenty of examples of micropeptides that have recently been discovered, although none of them have a strong cancer focus. Zhou 2024 [74] provides a robust insight into cancer-related peptides and their potential as anticancer therapeutics. However, what sets this review apart is its focus on multi-omics in micropeptide detection.

## 5. Conclusions

Recent breakthroughs in proteomics and genomics technologies have uncovered a hidden world of non-canonical micropeptides through various multi-omic approaches, including ribosome profiling, mass spectrometry, and machine learning. We emphasize the importance of a comprehensive multi-omic approach to fully elucidate the micropeptidome of cancer. Micropeptides have the potential to revolutionize cancer therapy by providing innovative, more effective, and less toxic options for its treatment, as well as advancing precision and personalized medicine. Further efforts in the research of non-canonical micropeptides will surely introduce a paradigm shift in proteomics, massively expanding our understanding of how many important protein species and proteoforms have flown under the radar because of the mis-annotation of their genome as non-coding.

## Figures and Tables

**Figure 1 proteomes-12-00026-f001:**
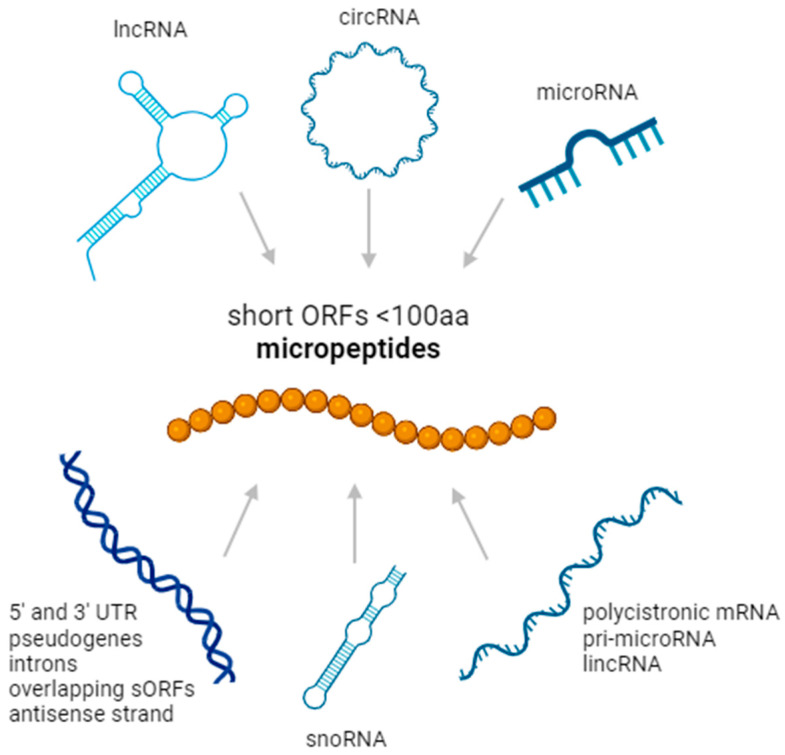
Origin of non-canonical micropeptides in the genome. Small peptides can be encoded by regions of the DNA such as 5′ and 3′ UTRs of coding regions, pseudogenes, intronic regions, sORFs overlapping with known coding regions, and the antisense strand of known coding regions. Known RNA sources of micropeptides are long non-coding RNA, circular RNA, microRNA, small nucleolar RNA, polycistronic mRNA, pri-microRNA, and long intronic non-coding RNA.

**Figure 2 proteomes-12-00026-f002:**
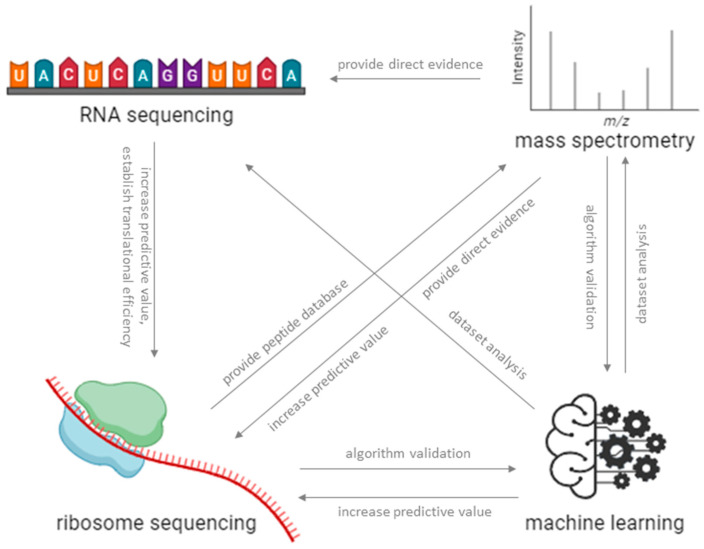
Integration of several OMICs methods, such as RNA sequencing, Ribo-Seq, MS, and machine learning, provides a clearer picture of the actual translatome complexity and is key to successful micropeptidome discovery.

**Table 3 proteomes-12-00026-t003:** Biological mechanisms and therapeutic implications of previously identified cancer-promoting micropeptides.

Micropeptide	Encoded by	Function	Mechanism	Cancer Type
ASAP [59]	lncRNA LINC00467	Drives colorectal cancer cell proliferation	Binds to ATP synthase, promotes activity	Colorectal cancer
SMIM30 [60]	lncRNA LINC00998	Drives cancer proliferation and growth	Interacts with SRC/YES1, activates MAPK signaling pathway	Hepatocellular carcinoma
STMP1 [61]	C7orf73	Promotes tumor cell migration	Interacts with DRP1 in mitochondria	Hepatocellular carcinoma
JunBP [62]	lincRNA LINC02551	Drives cancer metastasis	Binds to c-Jun, promotes activation	Hepatocellular carcinoma
XBP1SBM [63]	lncRNA MLLT4-AS1	Promotes growth, angiogenesis, and metastasis	Enhances VEGF expression	Breast cancer
ACLY-BP [64]	LINC00887	Drives cell proliferation	Enhances ACLY stability, increases acetyl-CoA production and lipid deposition	Renal cell carcinoma
APPLE [65]	lncRNA ASH1L-AS1	Initiates protein translation	Works within PABPC1 complex	Leukemic cancer
ASNSD1-uORF [66]	upstream ORF of ASNSD1	Necessary for cell survival	Interacts with prefoldin-like chaperone complex	Childhood medulloblastoma
DDUP [67]	lncRNA CTBP1-DT	Contributes to cisplatin resistance	Sustains DNA damage repair signaling	Ovarian cancer
PACMP [68]	lncRNA CTD-2256P15.2	Promotes growth, drives cancer therapy resistance	Inhibits CtIP-KLHL15 association, enhances PARP1-dependent poly(ADP-ribosyl)ation	Breast cancer
TRPC5OS [69]	antisense strand of TRPC5	Induces tumorigenesis	Interacts with ENO1 to increase glucose uptake	Breast cancer
pep-AKR1C2 [70]	exo-lncRNA lncAKR1C2	Promotes lymphangiogenesis and metastasis	Promotes expression of CPT1A by decreasing YAP phosphorylation	Metastatic gastric cancer
ATMLP [71]	lncRNA	Promotes tumorigenesis	Suppresses autolysosome formation through interaction with NIPSNAP1	Non-small cell lung cancer

## Data Availability

Not applicable.

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
