# Peer review of "Multi-Omic Approaches in Cancer-Related Micropeptide Identification"

_proteomes, 2024, doi:10.3390/proteomes12030026_

Round 1
Reviewer 1 Report
Comments and Suggestions for Authors
This submission and its objectives have good merit.
The authors have constructed a well-written review of known micropeptides in cancer with emphasis on analysing their roles in cancer progression or treatment, their potential as anti-cancer agents, and summarise principal omics methodology for micropeptide detection and identification. The review presents a balanced account of the prospects and potential of micropeptides alongside current limitations.
The authors have collated sufficient evidence, supported by appropriate citations, and with well-reasoned arguments, to make the convincing case that micropeptides are very worthy of investigation in cancer therapeutics.
Whilst overall, the references in support of statements in the narrative are sound, I recommend acknowledging the existence of other contemporary work in two recent reviews (below) that do delineate methods of micropeptide identification, detection and characterisation (in more detail). These are complementary rather than competitive works and what sets the current manuscript apart is its explicit cancer focus.
The role of micropeptides in biology | Cellular and Molecular Life Sciences (springer.com)
Vitorino, R., Guedes, S., Amado, F. et al. The role of micropeptides in biology. Cell. Mol. Life Sci. 78, 3285–3298 (2021). https://doi.org/10.1007/s00018-020-03740-3
Micropeptides Identified from Human Genomes (acs.org)
Jing Yuanyuan and Yin Xinqiang. Micropeptides Identified from Human Genomes. Journal of Proteome Research 2022 21 (4), 865-873 https://doi.org/10.1021/acs.jproteome.1c00889
Presentation
The manuscript is well proofread. I draw the authors’ attention to a minor anomaly between the abbreviations used in Figure 1 which describes micropeptides as < 100AA (upper case); whereas throughout the narrative lower case (aa) is used. Suggest modifying the Figure entry for consistency.
Content
For the greater part, the content is on-point, concise and well-written. I recommend the authors revisit the Discussion and Conclusions (sections 3. and 4.) because they are quite repetitive in some aspects e.g. statements around general properties of micropeptides (3. lines 288 – 292 and 4. Lines 340 – 342). I recommend resolving this issue and excising lines 344 – 347 (no need to remind the reader of what has been reviewed), concentrating statements into the Discussion and simplifying/shortening the Conclusions. I suggest the manuscript will benefit from the amendments to what is otherwise a very commendable review.
In summary
The manuscript content will be relevant and valuable for researchers at the entry level to the subject of biological micropeptides, and notably the oncology and drug discovery communities, as well as synthetic chemists working in the area of cancer therapeutics and requiring a biological perspective in drug design.
Author Response
Comment 1:
Whilst overall, the references in support of statements in the narrative are sound, I recommend acknowledging the existence of other contemporary work in two recent reviews (below) that do delineate methods of micropeptide identification, detection and characterisation (in more detail). These are complementary rather than competitive works and what sets the current manuscript apart is its explicit cancer focus.
The role of micropeptides in biology | Cellular and Molecular Life Sciences (springer.com)
Vitorino, R., Guedes, S., Amado, F. et al. The role of micropeptides in biology. Cell. Mol. Life Sci. 78, 3285–3298 (2021). https://doi.org/10.1007/s00018-020-03740-3
Micropeptides Identified from Human Genomes (acs.org)
Jing Yuanyuan and Yin Xinqiang. Micropeptides Identified from Human Genomes. Journal of Proteome Research 2022 21 (4), 865-873 https://doi.org/10.1021/acs.jproteome.1c00889
Reply 1:
We believe this review will benefit from the addition of this two articles as references, and have added them to the discussion part.
Comment 2:
The manuscript is well proofread. I draw the authors’ attention to a minor anomaly between the abbreviations used in Figure 1 which describes micropeptides as < 100AA (upper case); whereas throughout the narrative lower case (aa) is used. Suggest modifying the Figure entry for consistency.
Reply 2:
We thank the reviewer for pointing out the error, and have modified the figure accordingly.
Comment 3:
For the greater part, the content is on-point, concise and well-written. I recommend the authors revisit the Discussion and Conclusions (sections 3. and 4.) because they are quite repetitive in some aspects e.g. statements around general properties of micropeptides (3. lines 288 – 292 and 4. Lines 340 – 342). I recommend resolving this issue and excising lines 344 – 347 (no need to remind the reader of what has been reviewed), concentrating statements into the Discussion and simplifying/shortening the Conclusions. I suggest the manuscript will benefit from the amendments to what is otherwise a very commendable review.
Reply 3:
We appreciate the comments provided by the reviewer, and believe that they will strongly improve the manuscript. The discussion and conclusion part have now been amended according to the recommendations of the reviewer
comment 3:
The manuscript content will be relevant and valuable for researchers at the entry level to the subject of biological micropeptides, and notably the oncology and drug discovery communities, as well as synthetic chemists working in the area of cancer therapeutics and requiring a biological perspective in drug design.
Reviewer 2 Report
Comments and Suggestions for Authors
As small proteins with less than 100 amino acids, micropeptides produced from short open reading frames had been increasingly discovered and shown to play vital functional roles. The authors reviewed 16 micropeptides that are related to various cancers, in addition to multi-omic approaches for micropeptide identification and detection. However, the manuscript looks like half-baked with regard to the number of micropeptides mentioned and the number of cited references. The following concerns must be addressed before its acceptance.
1. There are a lot of reviews of micropeptides recently, such as https://doi.org/10.3389/fgene.2021.651485; https://doi.org/10.1631/jzus.B2300128; https://doi.org/10.1186/s12935-024-03281-w. These reviews covered a broad range of researches related to micropeptides, some of which was also mentioned in this manuscript. The authors should discuss how the current manuscript distinct to the existing reviews and justify the necessity and significance of this manuscript to the field.
2. From the title, it seems like a review on multi-omic approaches. Methods are only reviewed in section 2.1, which is a rather small part of the manuscript. A great part of the manuscript is left for the introduction of micropeptides in cancer in section 2.2, some of which were reported to “drive oncogenesis and cancer progression” instead of working against cancer. So, the title should be changed to be consistent with the content of the manuscript.
3. The last three methods listed in Table 1 seems not introduced in section 2.1 and should be also introduced with great details.
4. The section 2.1 may be moved out of Section 2, as the methods in section 2.1 are general methods to identify micropeptides, not the ones that are specific for micropeptides in cancer, not to mention the ones in anti-cancer therapy.
5. Only 16 micropeptides that are related to various cancers are reviewed in section 2.2 and listed in Table 2, which seems not consistent with the statement “Considering the large number of tumor-suppressing micropeptides listed in this review” in the discussion. This part is the major contribution to the field from my best understanding. However, the length of the part and the number of micropeptides is not enough for a publishable review article. The authors should either provide a comprehensive list or pay attentions to other recent advanced in the field of micropeptides and introduce them as well. Potential topic could be the functional micropeptides related to other human diseases other than cancer.
6. The origin of micropeptide is shown in Figure 1, however, a concise description should be also provided in the caption or in the text. It could be also interesting to introduce in detail the origin of micropeptides related to cancers.
7. In tables 1 and 2, references should be provided, for example, as a separate column.
8. The content of the first two paragraphs in the conclusion largely overlapped with the ones in the introduction part. These two paragraphs can be merged with the introduction. Thus, the conclusion part should be also revised to a great extent.
Author Response
Comment 1:There are a lot of reviews of micropeptides recently, such as https://doi.org/10.3389/fgene.2021.651485; https://doi.org/10.1631/jzus.B2300128; https://doi.org/10.1186/s12935-024-03281-w. These reviews covered a broad range of researches related to micropeptides, some of which was also mentioned in this manuscript. The authors should discuss how the current manuscript distinct to the existing reviews and justify the necessity and significance of this manuscript to the field.
Reply 1 :
We have now further differentiated our review from the recent ones, by explaining that we are focusing on technologies to detect cancer-related micropeptides. We believe this review is among the most comprehensive for the description of recently identified micropeptides, as well as multi-omic methods for their detection.
Comment 2: From the title, it seems like a review on multi-omic approaches. Methods are only reviewed in section 2.1, which is a rather small part of the manuscript. A great part of the manuscript is left for the introduction of micropeptides in cancer in section 2.2, some of which were reported to “drive oncogenesis and cancer progression” instead of working against cancer. So, the title should be changed to be consistent with the content of the manuscript.
Reply 2:
We have modified the title to be more in line with the topic of the manuscript.
Comment 3: The last three methods listed in Table 1 seems not introduced in section 2.1 and should be also introduced with great details.
Reply 3:
We thank the reviewer for pointing this out, and have introduced the methods as recommended.
Comment 4: The section 2.1 may be moved out of Section 2, as the methods in section 2.1 are general methods to identify micropeptides, not the ones that are specific for micropeptides in cancer, not to mention the ones in anti-cancer therapy.
Reply 4:
We have amended the structure of the manuscript to reflect this, splitting the main text of the manuscript into two sections, one of which focuses on micropeptide identification and one that discusses cancer-specific micropeptides.
Comment 5: Only 16 micropeptides that are related to various cancers are reviewed in section 2.2 and listed in Table 2, which seems not consistent with the statement “Considering the large number of tumor-suppressing micropeptides listed in this review” in the discussion. This part is the major contribution to the field from my best understanding. However, the length of the part and the number of micropeptides is not enough for a publishable review article. The authors should either provide a comprehensive list or pay attentions to other recent advanced in the field of micropeptides and introduce them as well. Potential topic could be the functional micropeptides related to other human diseases other than cancer.
Reply 5:
We have added eleven additional micropeptides to Section 3, splitting the section into two sub-sections for tumor-suppressing and tumor-promoting micropeptides. We believe that, in addition to the amount of micropeptides discussed, the manuscript’s major contribution is the discussion of multi-omic techniques for their detection. Furthermore, we believe that adding micropeptides related to other diseases would be out of scope, which is why we have decided to focus on cancer-specific ones.
Comment 6: The origin of micropeptide is shown in Figure 1, however, a concise description should be also provided in the caption or in the text. It could be also interesting to introduce in detail the origin of micropeptides related to cancers.
Reply 6:
We have expanded the Figure 1 legend as recommended.
Comment 7: In tables 1 and 2, references should be provided, for example, as a separate column.
Reply 7:
We thank the reviewer for pointing this out, and have provided references for the entries in the table.
Comment 8: The content of the first two paragraphs in the conclusion largely overlapped with the ones in the introduction part. These two paragraphs can be merged with the introduction. Thus, the conclusion part should be also revised to a great extent.
Reply 8:
We appreciate the reviewer’s comments. The Discussion and Conclusion part have been revised according to instructions
Round 2
Reviewer 2 Report
Comments and Suggestions for Authors
The authors have addressed all of the comments in the previous report.